# The Changes in Starch Gelatinization Behavior under the Influence of Acetic Acid in Vegetable Sponge Cake Batter in Order to Obtain New Snacks

**DOI:** 10.3390/polym14194053

**Published:** 2022-09-27

**Authors:** Manon Chemin, Olivier Paurd, Laure Villaceque, Alain Riaublanc, Patricia Le-Bail

**Affiliations:** 1INRAE, UR1268 Biopolymers Interactions Assemblies, Impasse Thérèse Bertrand-Fontaine, BP 71627, CEDEX 3, 44316 Nantes, France; 2Nbread Process, 3 Boulevard de l’industrie, 41700 Contres, France

**Keywords:** vegetable sponge cake, wheat flour, chickpea flour, quinoa flour, acetic acid, micro-differential scanning calorimetry

## Abstract

(1) Background: Adding white vinegar to the batter of a sponge cake without biological fermentation requires the effects of acidification on the batter to be checked, in particular concerning batter-to-crumb transition. (2) Methods: µDSC analyses were carried out on three batters formulated from flour, colza oil, salt, carrot, and water with or without the addition of white vinegar. (3) Results: Wheat, chickpea, and quinoa starches had gelatinization temperatures (T_Ge_) of 60.1, 72.4, and 70.5 °C at batter humidity and gelatinization enthalpies (ΔH_Ge_) of 9.2, 15, and 9.1 J/g_dry starch._ Due to the effect of the salt and carrot, the corresponding wholemeal batter had T_Ge_ of 64.2, 74.1, and 72.4 °C and ΔH_Ge_ of 10.5, 15.3, and 10.9 J/g_dry starch_. Acidified batters at pH 4 saw their T_Ge_ decrease, and their enthalpies increase compared to the controls. The calorimetric study of model mixtures revealed three different evolutions of ΔH_Ge_ as a function of pH, explained by the isoelectric behavior of flours and/or the attack of starch by acetic acid. (4) Conclusions: These results could be useful for adapting the cooking step of the acid batter in order to produce alternative snacks.

## 1. Introduction

Almost half (47%) of French people have an aperitif at least once a week [1]. With around 250 varieties of crunchable appetizers including crisps, tuiles, salted seeds, extrudates and biscuits, the appetizer snack sector is expected to grow by 4.5% between 2019 and 2024 globally [2]. The aperitif snack market has been evolving, integrating new production techniques, new flavors and bringing new societal, environmental, and nutritional dimensions to the products. In particular, the offer is adapting to new ways of consuming, by offering local, organic, gluten-free and up-cycling in the formulations [3]. Traditionally, crackers are made using a multi-step process: (1) fermentation of a dough made from wheat flour, (2) rolling the batter and cutting the pieces, (3) cooking [4]. Several effects of fermentation have been noted [5]: during this stage, the rheology of the dough evolves, allowing its lamination [6], aromatic molecules are formed, as well as acid molecules causing the decrease of pH.

In the context of our study, the use of an alternative process for manufacturing snacks, obtained by drying a sponge cake [7], does not involve a biological fermentation step. In order to compensate for this absence, the addition of white vinegar was considered. Indeed, beyond the sensory interest, the presence of vinegar in food formulations is interesting at the functional level. First, the acidic pHs caused by vinegar allow better preservation of products through their antibacterial effect [8,9,10] and can reach optimal values to reinforce the action of other raw materials, such as the activity of flour proteolytic enzymes [5], the biocidal effect of preservatives [11] and the antioxidant effect of certain ingredients [12]. The acidity induced by vinegar can also limit the formation of carcinogenic compounds, i.e., acrylamides, sometimes produced during the cooking of snacks [13]. Next, vinegar has certain health benefits, through its antibacterial and antioxidant activity [14]. Finally, the acetic acid of the vinegar can bring new properties to the mixtures by interacting with the other ingredients of the recipe. Indeed, many studies have been carried out on the use of different acids for the treatment of starch.

To our knowledge, acetic acid has been studied on wheat, potato, rice, and tapioca starch [15,16,17]. We want to provide new knowledge on the treatment of chickpea and quinoa flour with acetic acid. In the literature, there are also works relating to the study of the gelatinization of wheat, chickpea, and quinoa starches in the presence of HCL and H_2_SO_4_, but under lintnerization conditions [18,19,20]. Other studies, although fewer in number, have focused on the acidification of real batters but described only analysis of the acidification action on the finished product, for a formula of crackers with buckwheat [21], or formula of gluten-free bread [22].

One of the primary objectives of this invention is to produce high-fiber snacks using so-called “ugly” vegetables deemed unsuitable for sale, the batter studied contains a high content of vegetable fibers. For these snacks made using an innovative process, it is important to know the energy needed to allow complete gelatinization of the starch system in the batter during cooking. The objective of this work is therefore to study and compare the changes in gelatinization behavior under the influence of acetic acid of different starches (wheat, chickpea, and quinoa), present in acidified vegetable sponge cakes.

## 2. Materials and Methods

### 2.1. Materials

Organic T45 wheat flour (F_W_) (moisture 16.6%, starch 68.5% including 31.2% amylose and 68.8% amylopectin, protein 12%, fat 0.8%) was supplied by the Giraudineau mill (Saint Colomban, France). Desi variety organic chickpea flour (F_CP_ (moisture 10.9%, starch 40.9% including 32.4% amylose and 67.6% amylopectin, protein 16.8%, fat 6.3%) was provided by Qualisol (Castelsarrasin, France). Quinoa flour (F_Q_) (moisture 9.9%, starch 58.4% including 18.8% amylose and 81.2% amylopectin, protein 11.4%, fat) was obtained from seeds of Blond Berry Quinoa purchased from SAS Berry Graines (Nérondes, France). The unshelled seeds were crushed with a Retsch ZM200 brand ultracentrifugal grinder (Retsch, Haan, Germany) consisting of a 12-tooth rotor and a 0.2 mm mesh sieve. The carrot puree (C) (moisture 92.7%) was provided by Clément Faugier ETS (Privas, France) in frozen form. The puree pebbles were put in a cold room at 4 °C for 48 h before being used to defrost them. Rapeseed oil (O) (7.6% saturated fatty acids, 63.1% monounsaturated fatty acids, and 29.3% polyunsaturated fatty acids), fine table salt (Sa), and white vinegar (acetic acid, acidity 7.2%, pH 2.8) were bought in supermarkets (Nantes, France). The water (W) was millipore water, bubbled under N_2_ before use to overcome oxidation phenomena.

### 2.2. Sample Preparation

#### 2.2.1. Preparation of the Vegetable Sponge Cake Batter

The reference industrial sponge cake batter recipe had the particularity of being egg-free, gluten-free, yeast-free, and additive-free. The various reference recipes contain 30% of flour, 5% of oil, 0.3% of salt, 50% of carrot and 14.7% of added water. The formulations are written in dry basis as follows:F_a_ O_b_ Sa_c_ C_d_ W_y_

F: dry flour, O: rapeseed oil, Sa: salt, C: dry carrot, W: total water present in the mixture

a/b; a/c; and a/d: constant

a + b + c + d + y = 100

The different formulations are described in Table 1.

All the ingredients, balanced at room temperature, were weighed one by one then mixed in a cylindrical stainless-steel tank with a capacity of 2.5 L, using a mixing blade. The mixing speed was set at 150 revolutions per minute. The mixing time was 15 min for all batters.

#### 2.2.2. Preparation of Reference Model Mixtures

The reference model mixtures were flour and water (F_x_W_y_) primary model media. These simplified mixtures were made for two humidity values: the reference value (65%) and excess water (80%).

#### 2.2.3. Acidification with White Vinegar

New sponge cake batters were formulated at a target pH of 4 by replacing part of the water in the formulas with white vinegar (Table 2).

To simplify the coding in results and discussion parts, the proportion of each component in the mixture will no longer be specified.

### 2.3. pH Measurement

The pH of the batter was determined using a calibrated pH meter with a probe for pastes (Hanna instruments) at room temperature.

### 2.4. Zeta Potential of Flours

For each flour, a range of mixtures have been made in the presence of vinegar and water (55 mg of flour for 100 g of plasticizer) to reach 4 different pH values from 3 to 6. For each mixture, zeta potential has been measured by electrophoretic light scattering with a Zetasizer analyser (Malvern Panalytical, Malvern, UK). After equilibration during 20 min at 20 °C. The operation has been made on 5 samples per mixture

### 2.5. Micro Differential Scanning Calorimetry (µDSC)

The gelatinization properties of the starches were studied for all the mixtures with a µDSC 7 microcalorimeter (Setaram, Caluire, France). An empty cell was used as a reference. 300 to 500 mg of the sample were placed in a sealed µDSC Hastelloy capsule for thermal analysis. After equilibration at a temperature of 10 °C and maintaining at this temperature for 10 min, a rise from 10 to 120 °C at a rate of 1 °C/minute was carried out. The thermograms obtained were normalized in J/g of dry starch for analysis. The gelatinization peak temperature (T_Ge_) and the gelatinization enthalpy (ΔH_Ge_) were determined. In addition, the partial melting enthalpy was calculated from the start of the endotherm (in 1 °C increments) to plot the curve representing the cumulative enthalpy values as a function of temperature.

### 2.6. Statistical Analysis

For each formulation, 3 batters were produced, and 3 batter samples were taken per production. The mean and standard deviation were calculated, and a one-way ANOVA analysis was performed with Stat graphics software.

## 3. Results and Discussion

### 3.1. The pH of the Different Mixtures

Carrots benefit from a pH close to 6.0. This pH is relatively high compared to fruits such as orange or apple, which have pH values below 4.5, which eliminated an added effect not related to acidification. Table 3 brings together the pH values of the different mixtures.

### 3.2. Study of the Gelatinization of the Three Flours and Batters FOSaW and FOSaCW

Three different flours were used for this study. Wheat, a cereal flour traditionally used in bakery/pastry products and chickpea and quinoa, two alternative flours, one legume, the other pseudocereal, which have several industrial interests: on the one hand a nutritional aspect (gluten-free flours, but rich in proteins), on the other hand interesting features (emulsifying and foaming properties allowing them to replace egg white). Indeed, they had already given satisfactory results on dough rheology, bread texture and sensory analysis [23,24,25].

Wheat flour with excess water (80%) showed a gelatinization temperature T_Ge_ equal to 59.6 +/− 0.2 °C which was close to the results of Sopade et al. (60.3 °C) [26]. The measured enthalpy ΔH_Ge_ was 9.7 +/− 0.4 J/g_dry starch_ (9.2 +/− 0.2 J/g_dry starch_ for Sopade).

For chickpea flour, the gelatinization temperature and enthalpy found were T_Ge_ = 70.4 °C +/− 0.4 °C and ΔH_Ge_ = 12.6 J/g_dry starch_. In the literature, the authors mentioned slightly higher T_Ge_ gelatinization temperatures (70.6 to 73.3 °C) and much lower enthalpies (<5 J/g flour) [27,28,29]. However, the difference was explained by the fact that these authors did not relate the enthalpy to the quantity of dry starch. Quinoa flour had a gelatinization temperature TGe of 68.2 +/− 0.2 °C and a gelatinization enthalpy of 8.4 +/− 0.5 J/ g_dry starch_. These results were different from those mentioned in the literature [30,31], with lower gelatinization temperatures (64.5 °C and 67 °C, respectively) and generally higher enthalpies (16.8 J/g and 14.6 J/g, respectively). The differences observed between all the studies can be explained by the variety of composition (botanical origin, amount of amylose, amylopectin, starch in flour, and other constituents), differences in structure (organization of amylose and of amylopectin) [32], and the treatment of the flours used (cultivation conditions, milling, extraction, storage).

The T_Ge_ gelatinization temperatures of chickpea and quinoa flour were shifted to higher temperatures 72.4 +/− 0.1 °C and 70.5 +/− 0.1 °C, respectively, whereas they were stable for wheat starch when the water content decreased (65%). A more or less marked increase in T_Ge_ when the water content decreases has been observed for mixtures with chickpea flour [27] and also buckwheat [33], chestnut [34], and rice flour [35]. Moreover, as for wheat flour, a shoulder appeared on the Ge endotherm, which turned into a starch melting endotherm, M1 moving towards higher temperatures with decreasing water content. On the other hand, the enthalpies of gelatinization were not significantly different.

Batters FOSaW where F represented the flour (wheat or chickpea or quinoa), O the fat, S the salt, and W the total water of the system, had a higher gelatinization temperature (Figure 1).

This slight increase in the gelatinization temperature was linked to the presence of salt which competed with the starch for access to water. This competitive hydration resulted in less availability of water for starch gelatinization. D’Appolonia in 1972 and Spies and Hoseney in 1982 attributed the increase in gelatinization temperature to the solute’s ability to limit the amount of water available to starch, i.e., to reduce the water chemistry potential [36,37]. However, according to Wootton, the effect of salt on starch gelatinization was not solely due to the decrease in water availability since the energy of gelatinization did not decrease at the same rate as the amount of salt added to the aqueous phase [38]. The enthalpies of the FOSaW batters were not significantly modified compared to those obtained for F_35_W_65_ primary systems. This can be explained by the fact that the salt concentrations in our systems were very low (1.5 to 2% of the dry weight of the starch), which reduced its effect. Chiotelli and al [39] observed a decrease in the gelatinization enthalpy for wheat starch in presence of 16% of salt whereas there was no effect with low concentration of salt.

For the 3 complete batters FOSaCW where C represented the carrot, the differential enthalpy analysis study (Figure 1) showed that the addition of carrots to the batter led to a slight increase in temperature and in the gelatinization enthalpy for the three flours. For these systems, the majority of the water was provided by the carrot. The fiber content of the carrot was 2.7 g/100 g, the water provided by the carrot was therefore partly bound in the fibers and perhaps less accessible by the starch. However, at 65% water, the systems studied were still considered very close to excess water. The fibers, at this water content, therefore, did not cause any significant hindrance to gelatinization.

In the literature, fibers are added in the form of soluble or insoluble fibers. It was shown that insoluble or partially soluble fiber provided major effects on gelatinization (TGe decrease and ΔH increase). For example, Santos et al. [40] have studied the effects of flour replacement at different levels (6–34%) by soluble (inuline), partially soluble (sugar beet, pea cell wall), and insoluble (pea hull) fibers. Thermal profiles of wheat dough containing different fibers have been investigated by simulating baking, cooling, and storage in differential scanning calorimetry (DSC) pans. In general, dietary fiber incorporation into water-flour systems delayed endothermic transition temperatures for both gelatinization and retrogradation phenomena. Z. Goranova, et al. [41] evaluated the influence of functional ingredients on starch gelatinization in the sponge cake batter. They reduced the quantity of wheat flour in the sponge cake batter and added functional ingredients of 50% einkorn wholemeal flour, or 20% Jerusalem artichoke powder, or 35% cocoa husk powder. They showed that gelatinization occurs at higher temperatures and with higher energy consumption. They concluded that the retarding effect of the functional ingredients is related to the water binding capacity and the presence of dietary fiber.

We agreed with the literature but the water content of our sponge cakes being higher, the effect was less visible and for an equivalent water content our fiber content was lower.

### 3.3. Impact of Acetic Acid on the Six Batters FOSaW and FOSaCP

The study also focused on the impact of the acidification of a vegetable sponge cake batter with white vinegar on the thermal properties of the mixture. In order to determine the quantity of acetic acid to be used in the mixtures to reach a targeted pH of 4, a pH determination analysis was carried out on the batter with or without the presence of carrots.

A differential enthalpy analysis study was then conducted on FOSaCP batter to observe the hydrothermal properties of acidified batter. Gelatinization temperatures and enthalpies were shown in orange in Figure 1. A decrease in gelatinization temperature was observed when acetic acid was added and was much less marked for wheat than for chickpea and quinoa batters. Conversely, the gelatinization enthalpy increased for the three flours. Majzoobi and Beparva [15] observed the same effect on wheat starch treated with acetic acid for the T_Ge_ but the opposite effect for the gelatinization enthalpy and they suggested that the crystalline structure of the starch molecules inside the granules, was affected by the acids and hence less energy required to melt the crystalline structure. Likewise, Ohishi and collaborators [16] showed by DSC measurements that rice starch gelatinized more easily in the presence of acetic acid, they found that the addition of 0.2 M acetic acid resulted in a significantly lower gelatinization temperature (~1 °C) for rice starch. Hibi [42] also reported that the endothermic peak temperature of rice starch was lower in the presence of 0.33 M acetic acid. However, in both previous cases, the resultant pH of the heated starch dispersions was not reported by the authors. Villanueva et al. [17], for a pH adjusted at 4.5 did not observe any effect on gelatinization temperature for the rice starch dispersions, whereas the acidification of tapioca starch dispersions facilitated the gelatinization. All these authors agreed that the presence of proteins greatly influenced the gelatinization of these starches in the presence of acetic acid. Indeed, for Ohishi et al., the higher solubility and degradation of rice flour proteins in acetic acid than in distilled water might also accelerate the absorption, swelling, and gelatinization of starch of rice flour [16].

For other acids, the authors found an opposite effect to our results. On wheat starch in presence of HCl (T_Ge_ moved from 58.7 °C to 70.3 °C after 10.5 h [18]), chickpea wet starch with sulfuric acid (T_Ge_ moved from 132.8 °C to 139.6 °C [19]) or quinoa starch treated with H_2_SO_4_ (T_Ge_ moved from 66.7 °C to 86.4 °C [20]). Velasquez-Castillo and al, and Yixiang Xu and al also found an enthalpy increase [19,20]. These changes are attributed to the disappearance of the cooperative melting of the granules, which is facilitated by water uptake in the amorphous parts of the granule [18].

The whole of this work did not lead to the same conclusions, and the effects of acid on starch were sometimes contradictory between several works. The authors’ hypotheses expressed that gelatinization enthalpy value, related to the energy required to break down starch organization, could be dependent on crystalline organization, or water intake in amorphous area. It was difficult to compare all the results for two reasons. On the one hand, authors studied starch and not a whole batter and it was shown that ingredient addition could affect starch environment and might interact with acetic acid. Furthermore, the results have been obtained in different pH conditions and pH value could be an important factor to consider. For example, the impact of acetic acid on gelatinization could be related to the electrical potential of flours. Therefore, a study of F20P80 model systems was investigated as a function of pH by µDSC.

For the 3 flours, the T_Ge_ was stable up to a threshold pH (3.4; 4.2, and 4.0, respectively, for wheat, chickpea, and quinoa flour), then dropped when the pH decreased. Figure 2 showed that the starch gelatinization enthalpy in the F_w35_P_65_ standard mixtures was not influenced by the presence of acetic acid, whereas for the F_CP35_P_65_ standard mixtures, the evolution of ΔH_Ge_ followed a “bell-shaped” curve, presenting a maximum enthalpy of 17.6 +/− 1.3 J/g_dry starch_ at pH 3.8. Finally, for the F_Q35_P_65_ mixtures, the enthalpy ΔH_Ge_ increased significantly from 8.4 +/− 0.3 to 11.7 +/− 0.9 J/g_dry starch_ as soon as acetic acid was added, then stabilized.

The isoelectric points (pI) of the different flours were determined by measuring the zeta potential. Indeed, the zeta potential changes from positive to negative values when the pH increases and the pH at which the zeta potential is zero is the isoelectric point. The isoelectric points of the 3 flours (F_w_, F_CP,_ and F_Q_) were 5.8, 3.7, and 3.0, respectively. The increase in gelatinization enthalpy of F_CP_ flour may be related to the electrical point of chickpea starch. At pHi, the overall charge from surface proteins to starch was zero. If pH < pI, the overall charge was positive. If pH > pI, the overall charge was negative. It was around pH 4 (3.7) that the isoelectric potential of chickpea flour canceled out; this result was also shown by Emami in 2007 [43], leading to potential aggregation. A protein network around the chickpea starch could constitute a steric barrier to the entry of the plasticizer into the grain, which would therefore require more ΔH energy to gelatinize. These results agree with the hypothesis of Jacobs et al. (1998), which showed lowering the water uptake in the amorphous parts of the granule could increase the gelatinization enthalpy [18]. For wheat flour, the pHi was 5.8, therefore outside the range of pH studied, and for quinoa flour, the pHi was 3, which did not explain the increase in the gelatinization enthalpy at higher pHs. Another hypothesis would be that the action of acetic acid would reveal an attack on the wall of the starch granules.

In order to know the penetration rate of the plasticizer (water + acetic acid) into the starch granule, the curves of cumulative enthalpies were calculated and reported as a function of temperature. The partial gelatinization enthalpy curves for the three starches studied are shown in Figure 3. This study made it possible to observe the kinetics of loss of the ordered structure of starch at a temperature between 45 and 80 °C.

At free pH, (5.8; 5.2 and 6.4) the shapes of the curves illustrated for the F_35_W_65_ mixtures (Figure 3) were quite similar for the three flours F_w_, F_CP_, F_Q_. Indeed, the fusion of the ordered zones for F_CP_ and F_Q_ were very close to that of F_w_, contrary to that of tubers such as potato, which was much steeper, as reported in the literature (Waigh et al., 2000) [32]. As for the Fw, these observations indicated a de-structuring in two stages (loss of crystallinity and destruction of the helices) and can be partly explained by the surface state of the granules. Fannon et al. (1992) [44] showed that tuber granules appear smooth on the surface under scanning electron microscopy, whereas cereal starch granules such as those of wheat starch had pores of about 100 nm on their surface, generated at the time of granule biosynthesis and offering greater sensitivity to hydration. However, the temperatures at which the granular structure began to be destroyed were higher for chickpea and quinoa. As explained above, this may be due to the amylopectine structure [32], and the significant presence of proteins around the starch granules.

Although at pH 4 (P = water + acetic acid), the penetration rate of the plasticizer remained identical for the F_w35_P_65_ and F_CP35_P_65_ primary systems (Figure 4), a slight decrease in the slope of the curves were observed for the quinoa flour, which seemed to show that acetic acid attacked the surface of quinoa starch granules more easily. This is in accordance with Velasquez et al. who showed hydrogen bond modification in the amorphous area during quinoa starch acidic treatment [20]. This result explained the increase in the gelatinization enthalpy before a pH of 3 before passing the isoelectric point.

The total gelatinization enthalpy (ΔH) in the presence of acetic acid remained almost constant for the F_w35_P_65_ mixture but increased sharply for the F_CP35_P_65_ and F_Q35_P_65_ mixtures. Our results showed that two different mechanisms were at play to explain the increase in gelatinization enthalpy of the F_CP35_P_65_ and F_Q35_P_65_ mixtures. Indeed, for chickpea flour, at pH 4, the passage of the isoelectric point led to the aggregation of proteins creating a steric barrier to the plasticizer which caused an increase in enthalpy. For quinoa flour, the considerable increase in the total gelatinization enthalpy suggested a better organization of the ordered regions of the starch, linked to an attack of the amorphous regions by acetic acid. Indeed, according to Donovan’s [45] theory, acid hydrolysis preferentially attacks the amorphous regions in the granule, as a result, the crystallites are no longer destabilized by the amorphous parts.

## 4. Conclusions

In this study, we followed the changes in the gelatinization behavior of three starches (wheat, chickpea, and quinoa) under the influence of acetic acid in vegetable sponge cake in order to obtain new snacks.

The addition of vegetable fibers such as carrots to the sponge cake batter causes a very slight increase in the gelatinization temperature. The quantity of free water available, at this water content, is largely sufficient to allow complete gelatinization without modifying the cooking temperature. On the other hand, the acidification of the sponge cake with white vinegar to reach a pH of 4 causes a modification of its thermal properties. Indeed, the gelatinization of the starch appears at a lower temperature in the sponge cake batter formulated with chickpea and quinoa flour than the starch in the corresponding batter without vinegar. However, it is almost identical to the sponge cake made with wheat flour. This work will allow adjusting the microwave cooking parameters (time and power) to obtain complete gelatinization of these acidified vegetable sponge cakes.

Moreover, it is interesting to observe that the mechanisms of action of acetic acid on starch depend on the type of flour used. Indeed, two mechanisms of action of acetic acid on starch, one direct and the other indirect, have been demonstrated: (1) The acetic acid contained in vinegar acts directly on the starch of quinoa itself at low concentration with a marked increase in the speed of the plasticizer penetration into the granule. (2) For chickpea starch, acetic acid acts indirectly on its gelatinization by modifying the zeta potential of the other flour components, which probably causes steric hindrance for the supply of the plasticizer in the starch granules.

## Figures and Tables

**Figure 1 polymers-14-04053-f001:**
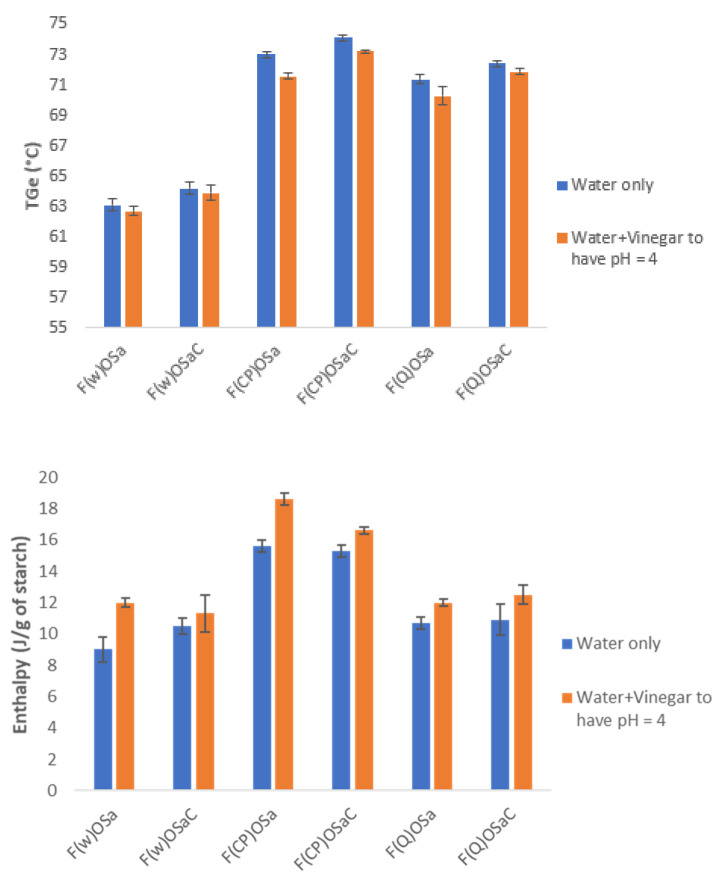
Gelatinization temperatures and enthalpies of different batters (with and without carrot and vinegar).

**Figure 2 polymers-14-04053-f002:**
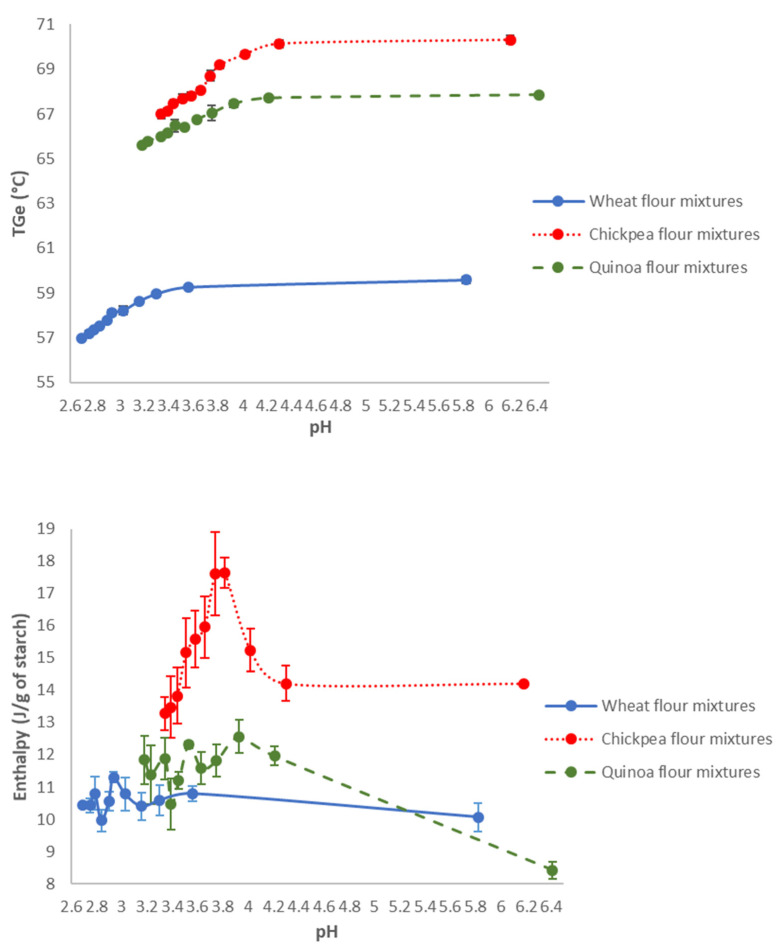
Gelatinization temperatures and enthalpies of standard mixtures as a function of pH.

**Figure 3 polymers-14-04053-f003:**
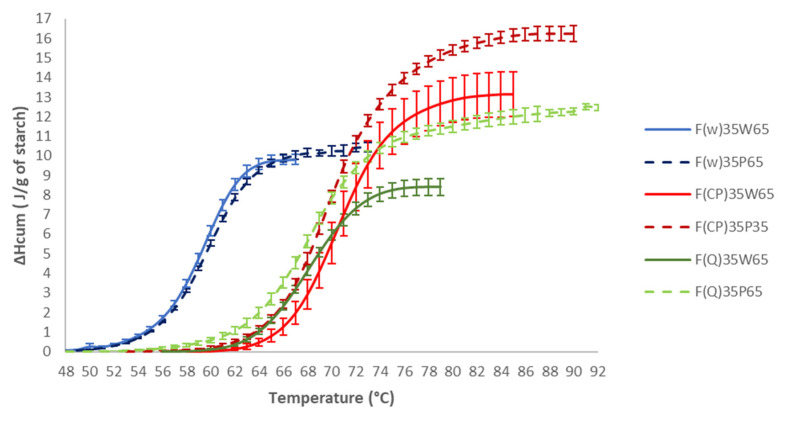
Effect of acetic acid on cumulated enthalpy for F_35_P_65_ mixtures.

**Figure 4 polymers-14-04053-f004:**
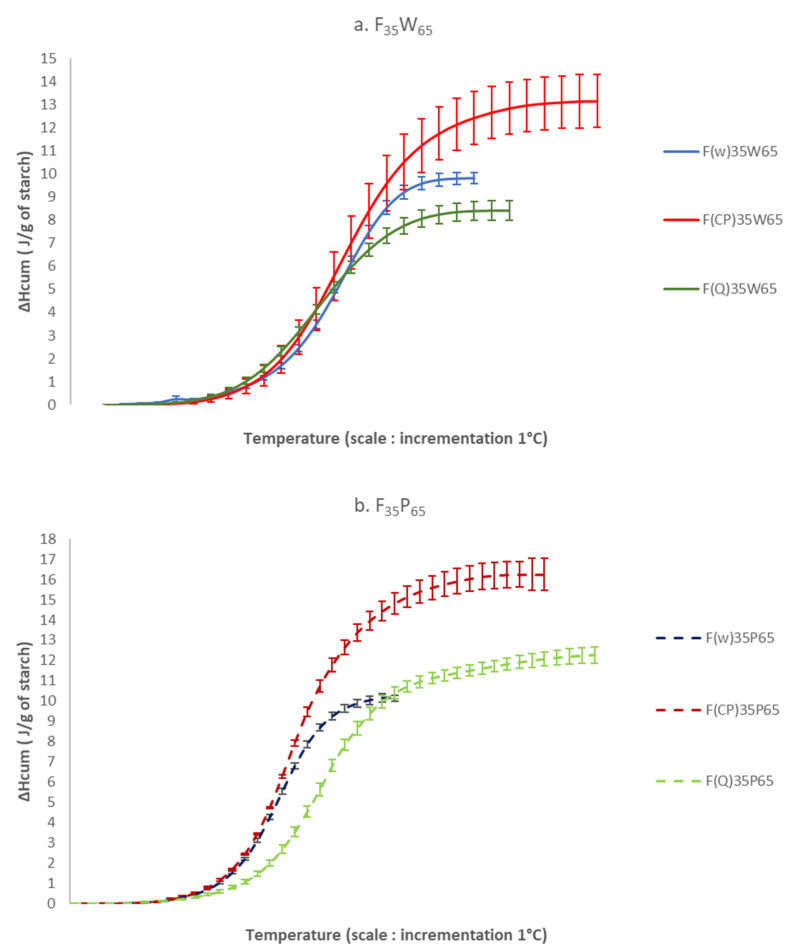
Superimpositions of the cumulative enthalpy curves for the F_35_W_65_ mixtures (**a**) and the F_35_P_65_ mixtures (**b**).

**Table 1 polymers-14-04053-t001:** Formulations of the different sponge cake batters with and without carrots.

Formulation	Dry Flour (g)	Oil (g)	Salt (g)	Dry Carrot (g)	Water (g)
F_w29_ O_5.6_ Sa_0.4_ W_65_	29	5.6	0.4	0	65
F_CP29_ O_5.6_ Sa_0.4_ W_65_	29	5.6	0.4	0	65
F_Q29_ O_5,6_ Sa_0.4_ W_65_	29	5.6	0.4	0	65
F_w26_ O_5_ Sa_0.4_ C_3.6_ W_65_	26	5	0.4	3.6	65
F_CP26_ O_5_ Sa_0.4_ C_3.6_ W_65_	26	5	0.4	3.6	65
F_Q26_ O_5_ Sa_0.4_ C _3.6_W_65_	26	5	0.4	3.6	65

**Table 2 polymers-14-04053-t002:** Formulations of the different acidified sponge cake batters with and without carrots.

Formulation	Dry Flour (g)	Oil (g)	Salt (g)	Dry Carrot (g)	Water (g)	White Vinegar (g)
F_w29_ O_5.6_ Sa_0.4_ P_65_	29	5.6	0.4	0	60.3	4.7
F_CP29_ O_5.6_ Sa_0.4_ P_65_	29	5.6	0.3	0	36.3	28.6
F_Q29_ O_5,6_ Sa_0.4_ P_65_	29	5.6	0.4	0	42.8	22.2
F_w26_ O_5_ Sa _0.4_C_3.6_ P_65_	26	5	0.4	3.6	54.1	10.9
F_CP26_ O_5_ Sa_0.4_ C _3.6_P_65_	26	5	0.4	3.6	45.5	19.5
F_Q26_ O_5_ Sa_0.4_C_3.6_ P_65_	26	5	0.4	3.6	45.5	19.5

**Table 3 polymers-14-04053-t003:** pH of the different sponge cake batters.

Mixtures	pH
F_w_OSaW	5.97 +/− 0.01
F_w_OSaCW	5.89 +/− 0.06
F_CP_OSaW	6.04 +/− 0.03
F_CP_OSaCW	5.83 +/− 0.09
F_Q_OSaW	6.36 +/− 0.02
F_Q_OSaCW	6.16 +/− 0.02

## Data Availability

Not applicable.

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
