# Peer review of "The Changes in Starch Gelatinization Behavior under the Influence of Acetic Acid in Vegetable Sponge Cake Batter in Order to Obtain New Snacks"

_polymers, 2022, doi:10.3390/polym14194053_

Round 1

Reviewer 1 Report

The document "impact of acetic acid on starch from a vegetable cracker batter" is a well presented and interesting document related to the behavior of different flour mixtures and acetic acid addition on cracker batter elaboration, more specifically in the starch gelatinization, although the experimental designs are simple, the information obtained it could be of interest on the subject, some minimal comments need to be addressed.

The title of the document does not reflect the importance of the study, authors need to improve it in order to highlight the importance of the paper.

The objective must be summarized to focused on the innovation of the work

Also the conclusions need to focus on the most important results, highlighting the novelty

Reviewer 2 Report

The manuscript has investigated the influence of vinegar on the thermal properties of vegetable cracker batters obtained from wheat, quinoa, or chickpea flour. The manuscript is poorly written and the experiments are very limited (only DSC test is performed). The title of the manuscript is not appropriate because starch is one of the components of flour and other components such as proteins may have significant effects on the obtained results. On the other hand, the results are not deeply discussed and there is not any evidence for the statements.

Other comments:

Line 66: Starch or flour?

Line 88: pH

Line 97 and Table 1: In line 97 you have use “V” for carrot, but in the table, you have used “C” for carrot.

Table 1 the second row: You have written “Sa0.3” but in the corresponding column you have written 0.4g salt.

Why the oil content of formulations is not the same?

Line 221: Delete “in”

Reviewer 3 Report

The importance of acidity for establishing rheological and textural properties of dough systems has been known for decades, especially for wheat and rye bread dough. Nevertheless, the role of starch properties is often neglected here, because of other flour components, such as proteins and arabinoxylans, which are often modified to a greater extent. Therefore, it seems entirely justified to study and compare the changes in starch gelatinization behavior under the influence of acetic acid, especially if the systems are designed for modern processing methods. The results presented by the authors show that such differences are decisive for calculating the energy necessary to heat the starch system in a way allowing its full gelatinization.

Nevertheless, the study needs significant changes before its publication. First of all the authors should reconsider the use of the word "cracker" for describing the potential product of the applied technology. In fact, the cited reference [4] clearly describes "crackers" in terms of their composition and production technology, and this description does not even loosely fit the method enclosed in the reference [7], which refers to "sponge cake" or "solid foam". The term "cracker batter" has no sense if we have in mind the process of dough lamination (key for cracker manufacture).

Secondly, the authors should redesign their sample coding. It is true that the applied system allows identifying individual formulations, but in a way very difficult for the reader. The coding should be as simple as possible and should include only the components that differ between samples (not salt and water). Instead of using numbers (3 digits), you could put the indices H, M, and L for the high, medium, and low content of the individual component, O would be better than FA for oil, W for wheat flour, and so on.

The aim of the study should be rephrased - isolated starch was not used in the study (which is a pity, because you can't be sure if the pH changes have a direct or indirect influence on starch properties). Also, the use of carrot puree should be clearly explained and included in the aim and conclusions (was it an example of vegetable fiber?).

The material and method section lacks information about pH measurement (maybe it would be also relevant to check or calculate the total titratable acidity of the batter, which would allow comparing the values with standard bread dough systems), as well as the determination of zeta-potential.

The discussion is very limited, more references should be found and discussed (e.g. references [15-17] are only mentioned in the introduction and not in the discussion). On the other hand, the references to cracker technology could safely be removed, together with a discussion on tuber starches (instead you may add a more detailed description of the botanical origin and chemical composition of studied flours).

Round 2

Reviewer 2 Report

The manuscript is not acceptable 

Reviewer 3 Report

I am not fully satisfied with the applied sample coding, but the amendments made by the authors seem to be sufficient.
